# A Fusion Positioning System Based on Camera and LiDAR for Unmanned Rollers in Tunnel Construction

**DOI:** 10.3390/s24134408

**Published:** 2024-07-07

**Authors:** Hao Huang, Yongbiao Hu, Xuebin Wang

**Affiliations:** National Engineering Laboratory for Highway Maintenance Equipment, Chang’an University, Xi’an 710064, China; 2018025006@chd.edu.cn (H.H.); hybiao@chd.edu.cn (Y.H.)

**Keywords:** unmanned roller, fusion positioning, loop closure detection, graph optimization

## Abstract

As an important vehicle in road construction, the unmanned roller is rapidly advancing in its autonomous compaction capabilities. To overcome the challenges of GNSS positioning failure during tunnel construction and diminished visual positioning accuracy under different illumination levels, we propose a feature-layer fusion positioning system based on a camera and LiDAR. This system integrates loop closure detection and LiDAR odometry into the visual odometry framework. Furthermore, recognizing the prevalence of similar scenes in tunnels, we innovatively combine loop closure detection with the compaction process of rollers in fixed areas, proposing a selection method for loop closure candidate frames based on the compaction process. Through on-site experiments, it is shown that this method not only enhances the accuracy of loop closure detection in similar environments but also reduces the runtime. Compared with visual systems, in static positioning tests, the longitudinal and lateral accuracy of the fusion system are improved by 12 mm and 11 mm, respectively. In straight-line compaction tests under different illumination levels, the average lateral error increases by 34.1% and 32.8%, respectively. In lane-changing compaction tests, this system enhances the positioning accuracy by 33% in dim environments, demonstrating the superior positioning accuracy of the fusion positioning system amid illumination changes in tunnels.

## 1. Introduction

The level of investment in large-scale projects, such as highways and airports, has been increasing in recent years. This trend has led to higher demands regarding the quality and efficiency of subgrade and pavement construction [1,2]. However, traditional manual driving rollers [3,4] have several shortcomings. The different skill levels of drivers may result in different compaction errors, leading to problems such as under-compaction or over-compaction [5,6], which can affect the construction quality. Additionally, accidents such as injuries to site personnel during the compaction process can occur. Therefore, traditional rollers cannot meet the demands of high quality and efficiency, nor can they guarantee the safety of personnel at the construction site.

With the rapid development of computer science, artificial intelligence, automation control and other technologies, unmanned rollers provide the possibility to solve the above problems [7,8]. To ensure higher accuracy than manual driving, unmanned rollers are designed to compact on the planned path through high-precision positioning. The compaction accuracy of unmanned rollers is directly affected by the accuracy of the positioning system.

Currently, unmanned rollers rely on a global navigation satellite system (GNSS) for positioning [5,6,7,9], offering high precision, all-weather capabilities and simple operation. However, a GNSS is only applicable in open areas. Due to the complexity and variability of the compaction scene, it is not possible to obtain GNSS signals in closed sites such as tunnels, as shown in Figure 1. Consequently, the roller cannot be turned on for autonomous compaction at this time.

In tunnel construction, researchers have proposed Simultaneous Localization and Mapping (SLAM) [10,11,12,13,14] as a replacement in order to achieve positioning and navigation in the absence of GNSS signals. By employing sensors such as cameras or LiDAR, the motion state of the roller is estimated and an environmental map is constructed in the absence of a priori environmental information. In our previous study [15], we employed camera-based visual odometry as a cost-effective alternative to LiDAR for the positioning of unmanned rollers in tunnels.

Unmanned rollers compact in a certain area for a long time, and there will be insufficient illumination in tunnels. A visual odometer will generate cumulative errors that cannot be corrected, resulting in poor positioning accuracy. Therefore, a feature layer fusion positioning system based on the camera and LiDAR for unmanned rollers is proposed. This system incorporates loop closure detection and LiDAR odometry in addition to visual odometry. Furthermore, in view of the prevalence of similar scenes in tunnels, this paper presents the innovative integration of loop closure detection with the compaction process of the roller in a fixed area and proposes a new method of selecting loop closure candidate frames.

This paper contributes in the following aspects. Firstly, the authors propose a GNSS-denied fusion positioning system based on the camera and LiDAR for unmanned rollers in tunnel construction. This system iteratively optimizes the residual error functions by integrating relative pose constraints between adjacent frames, loop closure constraints with historical frames and LiDAR prior pose constraints. Additionally, a new loop closure detection method based on the compaction process is introduced to address mismatches in similar scenes. Keyframes are intelligently grouped, and candidate loop closure frames are selected, demonstrating superior precision and real-time performance. Finally, the proposed fusion positioning system is compared with a traditional visual positioning system, evaluating the static positioning error, straight line positioning error and lane-changing error according to an unmanned roller test platform. The result indicates the effectiveness of the proposed method in reducing the cumulative error and offering good robustness under low-illumination conditions.

The structure of the paper is as follows. Section 2 reviews the related research on positioning methods for unmanned rollers. Section 3 proposes the principles of the feature layer fusion positioning system based on the camera and LiDAR and proposes an improved loop closure detection method. Section 4 analyzes the improvement in the precision and real-time performance of the improved loop closure detection method. The error of the fusion positioning system is compared with that of conventional visual positioning. Finally, Section 5 summarizes the conclusions and future work.

## 2. Related Work

Currently, most unmanned rollers use GNSS for high-precision positioning. Fang [16] relied on RTK positioning signals to achieve the automatic compaction of the roller. A path-following control model for a vibratory roller was established, which allowed the lateral error to reach 30 cm. Zhang [9] designed an unmanned roller system that relied entirely on the positioning accuracy of the GNSS. After analyzing the field test data, they found that the system improved the compaction quality and efficiency for earth and rock dams. Zhan [17] utilized the Attitude Heading Reference System (AHRS) to measure attitude information and correct the position and heading of the roller, which was previously measured by GPS only. This method reduced the positioning error by 0.197 m and the heading error by 1.6°. The aforementioned research was conducted in outdoor environments with good GNSS signals. For scenarios without GNSS, researchers have also proposed various positioning methods.

Song [18] introduced a hybrid positioning strategy that combines RFID technology and in-vehicle sensors. This strategy uses the received signal strength (RSS) from RFID tags and readers for preliminary positioning. However, it relies on extensive RFID infrastructure, which can be costly and complex to implement and maintain. Jiang [19] developed a tunnel vehicle localization method using virtual stations and reflections. The localization algorithms using filters improve the localization accuracy compared with a positioning algorithm without using filters.

Ultra-wide band (UWB) is one of the methods used to overcome the partial denial problem of the GNSS. Wang [20] combined UWB technology with an inertial navigation system (INS) to create a commonly used indoor positioning method. Gao [21] proposed an innovation gain-adaptive Kalman filter (IG-AKF) algorithm for rollers, which significantly improved the positioning accuracy compared to UWB and the standard KF.

Visual positioning is another method used to overcome the problem of partial denial of the GNSS. Sun [22] utilized image processing techniques to detect ground markings, resulting in the precise and real-time lateral positioning of the roller in the working area. However, this method only provides lateral positioning and does not achieve two-dimensional plane positioning. Mur-Artal et al. [23] proposed a camera-based ORB-SLAM2 system that uses the Oriented FAST and Rotated BRIEF (ORB) features and descriptors to accurately track the target’s positional coordinates. The system also includes a loop closure detection step to reduce errors and drift accumulation. Jakob Engel et al. [24] proposed direct sparse odometry (DSO), a visual odometry method based on sparse, direct structure and motion formulas. DSO suggests photometric calibration [25] to improve the robustness of monocular direct visual odometry. In our previous work [15], a vision-based odometry system for unmanned rollers was proposed, which used SURF and PnP to achieve two-dimensional positioning. However, this method is not suitable for environments with significant changes in illumination intensity.

In addition, a large number of researchers have begun to explore multi-sensor fusion positioning systems. Song [26] introduced PSMD-SLAM using panoptic segmentation and multi-sensor fusion to enhance the accuracy and robustness in dynamic environments. The primary drawback of PSMD-SLAM is its computational complexity, which may limit its real-time application in resource-constrained environments. Liu [27] discussed a multi-sensor fusion approach integrating the camera, LiDAR and IMU in outdoor scenes, highlighting time alignment modules and feature point depth recovery models for enhanced accuracy and robustness. Kumar [28] proposed a method for the estimation of the distances between a self-driving vehicle and various objects using the fusion of LiDAR and camera data, emphasizing low-level sensor fusion and geometrical transformations for accurate depth sensing.

The above-mentioned research covers different positioning methods in different scenarios. Among them, GNSS-based positioning methods are completely ineffective in tunnels. RFID technology and UWB-based positioning methods require the pre-deployment of equipment and precision measurement in construction sites, making their application inconvenient. Visual-based positioning methods are susceptible to changes in illumination intensity, leading to decreased positioning accuracy. Therefore, this paper proposes a tunnel fusion positioning system based on the camera and LiDAR for unmanned rollers. It incorporates loop closure detection and LiDAR odometry as supplements to visual odometry, reducing the impact of illumination variations. Additionally, a loop closure detection method based on the compaction process is proposed, enabling the high-precision identification of loop closure frames. Ultimately, this achieves high-precision positioning and navigation for unmanned roller in tunnels.

## 3. Methods

### 3.1. Composition of Fusion Positioning System

The system uses a camera and a LiDAR as inputs, and the framework is shown in Figure 2. It consists of five modules, and the principles and roles of each module are described in detail in the following.

External parameter calibration. This paper integrates two sensors, a camera and a LiDAR, which are not initially aligned in a unified coordinate system. Therefore, external parameter calibration is essential to unify them under the coordinate system of the roller.Visual odometry. It processes raw and depth images from the camera and conducts feature extraction and matching to derive matching point pairs between adjacent frames [29,30,31,32]. Subsequently, pose constraints are solved based on the matching point pairs. Visual odometry has been extensively discussed in our previous research, as shown in Figure 3.Loop closure detection. The aim is to determine whether the current position forms a loop closure frame with any historical positions. When a loop closure is detected, pose constraints are computed to rectify the accumulated errors.LiDAR odometry. LiDAR odometry is achieved through LiDAR point cloud matching. The resulting output serves as prior pose constraints inputted into the fusion positioning system.Feature layer fusion module. This module merges the pose constraints from the visual odometry, loop closure detection and LiDAR odometry modules into a unified framework based on graph optimization. By minimizing the reprojection errors, the fusion module enables the high-precision positioning of the unmanned roller.The aim of all sub-modules is to construct a fusion positioning model encompassing the LiDAR point clouds and texture/color data from the camera. This integration is critical for unmanned rollers in addressing the decreased positioning precision resulting from inadequate illumination in tunnel construction.

### 3.2. Loop Closure Detection

Due to the abundance of visually similar scenes encountered during tunnel construction, as shown in Figure 4, the issue of false positives in loop closure detection arises. Despite their visual resemblance, they do not constitute loop closure frames. Therefore, this paper introduces a new method that integrates the compaction process of rollers to refine the selection of loop closure candidate frames. Moreover, employing the visual bag-of-words (BoW) model [33], similarity detection is implemented to identify potential loop closure frames among the candidates. Ultimately, the validity of the loop closure frames is confirmed through a reprojection error analysis, ensuring accurate identification.

#### 3.2.1. Candidate Loop Closure Frames Based on Compaction Process

As shown in Figure 5a, the compaction process of the roller within the tunnel unfolds through several sequential stages. Initially, the roller awaits deployment at the starting point within the compaction zone, during which pertinent compaction parameters are defined. These parameters encompass the lateral width w, forward distance d, rolling speed v and number of compaction passes p. Subsequently, the roller commences the compaction of the first lane from the starting position, advancing forward and subsequently reversing the compaction direction upon reaching the predetermined forward distance. With the completion of the first lane compaction task, the roller executes a lane change to the right, initiating the compaction of the second lane. It systematically proceeds with lane shifting from the starting point to the endpoint directionally, culminating in the comprehensive compaction of the entire tunnel area. Therefore, the roller follows a fixed path within each lane, with loop closure occurring only when there is a change in the direction of motion. The proposed loop closure detection process is depicted in Figure 5b.

Due to the substantial overlap of information between adjacent image frames, including all frames in the loop closure detection process would not only increase the likelihood of false positives but also waste computational resources. Therefore, the first step is to extract keyframes. A new keyframe must be a certain number of frames (*n_min_*) away from the previous keyframe to ensure less overlap in the field of view. Additionally, the selected frame must have a sufficient number of extracted feature points. Furthermore, a new keyframe must be inserted whenever there is a significant pose change, to prevent abrupt trajectory shifts and ensure stable tracking by the positioning system.

After extracting the keyframes, the compaction direction must first be determined, i.e., whether *v* is greater than 0. When *v* > 0, the keyframes extracted during forward compaction are saved in set $KF_{f}$. When *v* < 0, the keyframes from backward compaction are saved in set $KF_{b}$. Keyframes within the same set are not subjected to loop closure detection against each other. Loop closure detection between keyframes is performed only when there is a change in the direction of movement.

When the roller changes from forward to backward, the keyframes in set $KF_{f}$ are temporarily fixed (denoted as $KF_{f}=\{KF_{f_{1}},KF_{f_{2}},\cdots KF_{f_{n1}}\}$), and the current keyframe $KF_{b_{x}}$ is searched for the candidates of loop closure within the search area in reverse order, while simultaneously being added to set $KF_{b}$. Conversely, when the roller changes from backward to forward, the keyframes in set $KF_{b}$ are temporarily fixed (denoted as $KF_{b}=\{KF_{b_{1}},KF_{b_{2}},\cdots KF_{b_{n}}\}$), and the current keyframe $KF_{f_{n+x}}$ is searched for loop closure within the search area in reverse order within set $KF_{b}$ and then added to set $KF_{f}$.

The length threshold *L* for the search area is adjusted according to Equation (1):(1)$$N_{KF_{x}}-\mathrm{int}\left(\frac{T_{KF_{n}}-T_{KF_{1}}}{T_{KF_{x}}}\right)\leq L\leq N_{KF_{x}}+\mathrm{int}\left(\frac{T_{KF_{n}}-T_{KF_{1}}}{T_{KF_{x}}}\right)$$
where $N_{KF_{x}}$ represents the position of the current keyframe in the keyframe set, $T_{KF_{n}}-T_{KF_{1}}$ denotes the total time taken to extract the keyframe set when the direction of speed changes and $T_{KF_{x}}$ indicates the time elapsed from the change in speed direction to the current keyframe. Finally, a set of candidate loop closure frames $KF_{CL}$ is obtained.

#### 3.2.2. Similarity Detection

The similarity between the current keyframe and the candidate loop closure frames in the set $KF_{CL}$ is assessed, and the frame with the highest similarity is selected as the loop closure frame. In this study, the SURF features from visual odometry are clustered using the K-means method to generate word vectors composed of ID numbers and weights. These vectors are organized into a dictionary using a k-d tree structure. Given the image features, the corresponding words can be retrieved from the dictionary.

After obtaining *N* features and their corresponding words for a frame, a distribution histogram representing the image in the dictionary is constructed. The Term Frequency–Inverse Document Frequency (TF–IDF) method is employed to assign weight coefficients, reflecting the importance of different words in distinguishing features. TF represents the frequency of a word’s occurrence in a single image; the higher the frequency, the greater the discriminative power. IDF indicates the frequency of a word’s occurrence in the dictionary; the lower the frequency, the greater the discriminative power.

For two keyframes, $KF_{n}$ and $KF_{m}$, their corresponding bag-of-words vectors are denoted as $v_{KF_{n}}$ and $v_{KF_{m}}$ respectively. The similarity between these vectors is measured according to Equation (2).
(2)$$s(v_{KF_{n}}-v_{KF_{m}})=1-\frac{1}{2}\left|\frac{v_{KF_{n}}}{\left|v_{KF_{n}}\right|}-\frac{v_{KF_{m}}}{\left|v_{KF_{m}}\right|}\right|=\frac{1}{2}\sum_{i=1}^{N}\left(\left|v_{KF_{n}i}\right|+\left|v_{KF_{m}i}\right|-\left|v_{KF_{n}i}-v_{KF_{m}i}\right|\right)$$

The resulting similarity scores fall within the range [0, 1] and are sorted in descending order. The candidate frame with the highest similarity is selected as the final loop closure frame. Subsequently, the relative pose between the two frames is computed and employed as the loop closure constraint in the fusion positioning system for unmanned rollers.

### 3.3. LiDAR Odometry

To mitigate the impact of different illumination levels on the positioning accuracy in tunnels, the fusion positioning system for unmanned rollers integrates LiDAR odometry based on LiDAR, in addition to visual odometry and loop closure detection, thereby establishing corresponding prior pose constraints.

Let R(t,α) represent the LiDAR scan function, where *t* denotes the time and α denotes the coordinates of the scan points. The polar coordinates of point *P* in the LiDAR’s coordinate system are denoted as P(r,θ), with the coordinates expressed as in Equation (4), where *FOV* represents the scanning angle of the LiDAR and *N* represents the number of LiDAR scan points.
(3)$$\alpha =\frac{N-1}{FOV}\theta =k_{a}\theta$$

$P^{\prime }$ represents the scan point at a time interval Δt between consecutive scans. Then, the scan function at any point in the second scan can be approximated using Taylor expansion as follows [34]:(4)$$R(t+\Delta t,\alpha +\Delta \alpha )=R(t,\alpha )+\frac{\partial R}{\partial t}(t,\alpha )\Delta t+\frac{\partial R}{\partial \alpha }(t,\alpha )\Delta \alpha +O(\Delta t^{2},\Delta \alpha^{2})$$

By neglecting higher-order terms, when the scan range and the coordinates of points change within [t,t+Δt], the gradient of the scan function is approximately
(5)$$\frac{\Delta R}{\Delta t}\approx R_{t}+R_{\alpha }\frac{\Delta \alpha }{\Delta t}=R_{t}+R_{\alpha }k_{a}\theta^{\prime }$$

Equation (6) represents the range flow constraint equation, with $R_{t}=\frac{\partial R}{\partial t}(t,\alpha )$, $R_{\alpha }=\frac{\partial R}{\partial \alpha }(t,\alpha )$ and ΔR=R(t+Δt,α+Δα)−R(t,α).

To express the velocity of all points within the scan range, the velocity (r,θ) is rewritten in the Cartesian coordinate system of the LiDAR:(6)$$\left\{ \begin{array}{l} r^{\prime }=x^{\prime }\cos\theta +y^{\prime }\sin\theta \\ r\theta^{\prime }=y^{\prime }\cos\theta -x^{\prime }\sin\theta \end{array}\right.$$

Assuming that the environment consists of static rigid bodies, the motion of all scan points is attributed to the intrinsic motion of the LiDAR. Hence, the velocity of the LiDAR and the velocity of the scan points possess the same value but the opposite direction.
(7)$$\left(\begin{array}{c} x^{\prime }\\ y^{\prime } \end{array}\right)=\left(\begin{array}{c} -v_{x,s}+y\omega_{s}\\ -v_{y,s}-x\omega_{s} \end{array}\right)$$

Let $\xi_{s}=(v_{x,s},v_{y,s},\omega_{s})$ denote the sensor velocity and (x,y) denote the Cartesian coordinates of point *P*. By substituting Equation (7) into Equation (6) and applying the rigid assumption given in Equation (8), the range flow constraint equation can be transformed into a LiDAR velocity constraint:(8)$$\left(\cos\theta +\frac{R_{\alpha }k_{\alpha }\sin\theta }{r}\right)v_{x,s}+\left(\sin\theta -\frac{R_{\alpha }k_{\alpha }\cos\theta }{r}\right)v_{y,s}+\left(x\sin\theta -y\cos\theta -R_{\alpha }k_{\alpha }\right)\omega_{s}+R_{t}=0$$

Each scan point imposes restrictions on sensor motion. By substituting the angle and coordinates of each scan point into Equation (9), the velocity and pose constraints of the LiDAR can be determined.

### 3.4. Feature Layer Fusion Model Based on Graph Optimization

Figure 6 illustrates the feature layer fusion model based on graph optimization. Triangles denote the keyframe nodes of the visual odometry, while squares represent the nodes of the LiDAR odometry. Blue lines indicate the standard pose constraints between keyframes, red lines signify the loop closure pose constraints formed when a loop closure keyframe is identified and black lines depict the prior pose constraints from the LiDAR odometry.

In the fusion positioning system for unmanned rollers, all observations and states are jointly optimized, with the residual errors of each constraint assigned weights, referred to as the information matrix. Considering the information matrix, the total residual error can be expressed as
(9)$$F(x)=\sum_{\langle i,j\rangle \in C}e_{ij}^{T}\Omega_{ij}e_{ij}$$

The optimization problem can then be formulated as
(10)$$x^{*}=\arg\min F(x)$$

There are three types of residual error: the pose residual error between adjacent frames, the relative pose residual error of the loop closure frames and the prior pose residual error from the LiDAR odometry. Since the first two constraints are based on relative poses, the forms and derivations are identical. However, the prior pose constraint of LiDAR odometry pertains to single-frame observations. Consequently, the residual error model is categorized into two types.

#### 3.4.1. Residual Error Model of Relative Pose

In graph optimization, the node represents the pose of the sensor, denoted by $\xi_{1},\xi_{2}\cdots \xi_{m}$. The relative motion between nodes $\xi_{i}$ and $\xi_{j}$, denoted by $\Delta \xi_{ij}$, can be expressed as in Equation (12):(11)$$\Delta \xi_{ij}=\xi_{i}^{-1}\circ \xi_{j}=\ln{\left(\exp({\left(-\xi_{i}\right)}^{\wedge })\exp({\xi_{j}}^{\wedge })\right)}^{\vee }$$

When there is a pose error, the residual error is calculated using Equation (13):(12)$$e_{ij}=\ln{\left(T_{ij}^{-1}T_{i}^{-1}T_{j}\right)}^{\vee }=\ln{\left(\exp({\left(-\xi_{ij}\right)}^{\wedge })\exp({\left(-\xi_{i}\right)}^{\wedge })\exp({\xi_{j}}^{\wedge })\right)}^{\vee }$$

Two variables, $\xi_{i}$ and $\xi_{j}$, need to be optimized. A left perturbation is added to both $\xi_{i}$ and $\xi_{j}$, resulting in $\delta \xi_{i}$ and $\delta \xi_{j}$, which can be expressed as
(13)$${\hat{e}}_{ij}=\ln{\left(T_{ij}^{-1}T_{i}^{-1}\exp({\left(-\delta \xi_{i}\right)}^{\wedge })\exp(\delta {\xi_{j}}^{\wedge })T_{j}\right)}^{\vee }$$

Equation (14) is simplified:(14)$$\begin{array}{ll} {\hat{e}}_{ij} & =\ln(T_{ij}^{-1}T_{i}^{-1}\exp((-\delta \xi_{i}{)}^{\wedge })T_{j}\exp((Ad(T_{j}^{-1})\delta \xi_{j}{)}^{\wedge }{)}^{\vee }\\ & \approx e_{ij}-J^{-1}(e_{ij})Ad(T_{j}^{-1})\delta \xi_{i}+J^{-1}(e_{ij})Ad(T_{j}^{-1})\delta \xi_{j} \end{array}$$

The Jacobian matrix of the residual error in Equation (15) with respect to $T_{i}$, $T_{j}$ is, respectively,
(15)$$\begin{array}{cc} A_{ij}=\frac{\partial e_{ij}}{\partial \delta \xi_{i}}=-J^{-1}(e_{ij})Ad(T_{j}^{-1}) & B_{ij}=\frac{\partial e_{ij}}{\partial \delta \xi_{j}}=J^{-1}(e_{ij})Ad(T_{j}^{-1}) \end{array}$$
(16)$$J^{-1}(e_{ij})\approx I+\frac{1}{2}\left[\begin{array}{cc} \phi_{e}^{\wedge } & \rho_{e}^{\wedge }\\ 0 & \phi_{e}^{\wedge } \end{array}\right]$$

Perform first-order Taylor expansion on the residual error and find the Jacobian matrix of the residual error with respect to the pose $J_{ij}$:(17)$$e_{ij}(x_{i}+\Delta x_{i},x_{j}+\Delta x_{j})=e_{ij}(x+\Delta x)\approx e_{ij}+J_{ij}\Delta x$$

For each residual error block, there is
(18)$$\begin{array}{ll} F_{ij}(x+\Delta x) & =e_{ij}(x+\Delta x{)}^{T}\Omega_{ij}e_{ij}(x+\Delta x)\approx (e_{ij}+J_{ij}\Delta x{)}^{T}\Omega_{ij}(e_{ij}+J_{ij}\Delta x)\\ & ={e_{ij}}^{T}\Omega_{ij}e_{ij}+2{e_{ij}}^{T}\Omega_{ij}J_{ij}\Delta x+\Delta x^{T}{J_{ij}}^{T}\Omega_{ij}J_{ij}\Delta x\\ & =c_{ij}+2{b_{ij}}^{T}\Delta x+\Delta x^{T}H_{ij}\Delta x \end{array}$$

At this point, the Gaussian–Newton method is used to optimize the solution.

#### 3.4.2. Residual Error Model of LiDAR Prior Pose

The LiDAR observation is a unitary edge. Unlike visual odometry and loop closure, which connect two pose states, this observation connects only one pose state. It directly provides the observed value of the state quantity, and its corresponding residual error is the difference between the observed value and the state quantity.
(19)$$e_{i}=\ln{\left(Z_{i}^{-1}T_{i}\right)}^{\vee }=\ln{\left(\exp{\left(-\xi_{zi}\right)}^{\wedge }\exp(\xi_{i}^{\wedge })\right)}^{\vee }$$

By adding perturbations to the residual error, it can be concluded that
(20)$$\begin{array}{ll} {\hat{e}}_{i} & =\ln(Z_{i}^{-1}\exp(\delta \xi_{i}^{\wedge })T_{i}{)}^{\vee }{\hat{e}}_{i}=\ln{\left(Z_{i}^{-1}T_{i}\exp({\left(Ad(T_{i}^{-1})\delta \xi_{i}\right)}^{\wedge })\right)}^{\vee }\\ & =\ln{\left(\exp(e_{i}^{\wedge })\exp({\left(Ad(T_{i}^{-1})\delta \xi_{i}\right)}^{\wedge })\right)}^{\vee }\approx e_{i}+{J_{r}}^{-1}(e_{i})Ad(T_{i}^{-1})\delta \xi_{i} \end{array}$$

Therefore, the Jacobian of the residual error with respect to $T_{i}$ is
(21)$$\frac{\partial e_{i}}{\partial \delta \xi_{i}}={J_{r}}^{-1}(e_{i})Ad(T_{i}^{-1})$$

The subsequent process is consistent with that in Section 3.4.1. At this point, the Gaussian–Newton method is also used to optimize the solution.

## 4. Results and Discussion

Figure 7 shows the test platform, which consists of a double-drum roller, a stereo camera, a LiDAR, a controller and an embedded computer. The double-drum roller’s forward and backward movement is controlled by adjusting the inlet and outlet oil volumes of the hydraulic pump through the proportional solenoid valve. The steering angle is controlled by adjusting the oil volume of the steering cylinder. Table 1 shows the parameters of the double-drum roller.

### 4.1. Results for Loop Closure Detection

Traditional loop closure detection methods are based on the DBoW3 library, which involves extracting keypoints and descriptors. In this experiment, to eliminate the influence of extraneous factors, SURF operators are similarly employed. Subsequently, a visual vocabulary is created, and the features of each image are transformed into a BoW histogram. The BoW histogram of the current image is compared with those stored in the database, and potential loop closure frames are identified through a comparison scoring function. The new loop closure detection method proposed for the fusion positioning system of an unmanned roller improves upon this process by incorporating a selection and classification process for keyframes, achieving more efficient detection within a specified range. The main evaluation metrics are the precision under different lighting conditions and the real-time performance.

#### 4.1.1. Precision

The precision of loop closure detection refers to the probability of correctly identifying a loop closure when the roller passes through similar scenes [35], typically described using *Precision* and *Recall*. It is defined by four statistical measures: true positive (*TP*) represents the number of correctly detected loop closures; true negative (*TN*) represents the number of correctly rejected loop closures; false positive (*FP*) represents the number of incorrectly identified loop closures; and false negative (*FN*) represents the number of loop closures incorrectly rejected. The formulas for *Precision* and *Recall* are as follows:(22)$$\begin{array}{cc} Precision=\frac{TP}{TP+FP} & Recall=\frac{TP}{TP+FN} \end{array}$$

To validate the recognition accuracy and robustness to illumination changes of the improved loop closure detection method based on the compaction process, precision–recall experiments were conducted during the first compaction under illumination intensities of 100 lux and 22 lux, respectively. The results are presented in Figure 8.

At an illumination intensity of 100 lux, with a recall rate of 60%, the precision rates are 99.3% and 97.1%. When the recall rate is 80%, the precision rates are 61.4% and 56.3%. Compared to the traditional loop closure detection method based on DBoW3, the improved method increases the precision by 5.1%, enhancing the recognition accuracy of loop closure frames and effectively filtering out a significant number of false positives in repetitive tunnel scenes.

Additionally, by comparing the left and right panels of Figure 8, it can be seen that, under different illumination levels, the precision at an 80% recall rate is 61.4% and 55.3%, respectively. This slight decrease in precision under lower illumination is due to the loop closure detection being based on visual features, which are more difficult to capture clearly in low-light conditions. However, the overall analysis indicates that this method still exhibits good robustness to changes in illumination.

#### 4.1.2. Real-Time Performance

Real-time performance is another crucial metric [36], where the time taken for loop closure detection must be significantly shorter than the data update cycle to meet the real-time requirements of the positioning system. The factors influencing the real-time performance include not only the complexity of the algorithm but also the computational capabilities of the hardware platform on which the algorithm operates, as well as software-level data scheduling and management.

The roller performs compaction in both the forward and reverse directions. The computation times for all keyframes using different loop closure detection methods are compared in Figure 9. When the direction of the roller remains unchanged, the proposed method eliminates all false candidate frames, thereby avoiding subsequent similarity checks. The average keyframe processing time for the traditional method is 7.89 ms, whereas it is 4.93 ms for the proposed method, thereby reducing the computational time for loop closure detection within the fusion positioning system.

#### 4.1.3. Positioning Experiment for Improved Loop Closure Detection Method

This section validates the improvement in positioning accuracy achieved by the improved loop closure detection method. Figure 10a,b display the trajectories and errors of the roller using different loop closure detection methods. The improved method reduces the median lateral error from 8.5 cm to 7.3 cm. Additionally, when compared to the ORB with the traditional loop closure detection method, the median lateral error decreases by 2.2 cm. This demonstrates that correctly eliminating the false loop closure frame in similar scenes allows for accurate pose association between the current frame and historical frames, thereby minimizing the impact of accumulated errors on the positioning accuracy.

In summary, the improved loop closure detection method significantly enhances both the precision and real-time performance compared to traditional methods. Furthermore, it exhibits robust performance under different illumination levels, ultimately reducing the positioning errors compared to systems using traditional loop detection methods. This effectively mitigates the cumulative errors encountered by the roller during forward and backward compaction.

### 4.2. Static Positioning Test

We chose ORB-SLAM2 as the comparative method in the static and dynamic positioning experiments for the following reasons. (1) Performance Benchmarking: ORB-SLAM2 is widely recognized as a benchmark in the field of SLAM due to its robustness and efficiency. Its widespread use in various research studies allows for a meaningful comparative analysis. (2) Feature Efficiency: One of the key strengths of ORB-SLAM2 is its efficient use of ORB features, which are highly effective in environments lacking GNSS signals, as is the focus of our study. (3) Availability and Accessibility: ORB-SLAM2 is open-source and has well-documented implementations, making it accessible for comparative evaluations.

Since the unmanned rollers need to remain stationary before starting the compaction process to receive the compaction parameters and task assignments, it is crucial that the positioning system does not drift during this period. Therefore, a static positioning test was conducted. The roller was kept stationary, and the relevant methods were executed to obtain the static positioning error, as shown in Figure 11. 

After removing outliers, the proposed fusion system achieved an average and maximum static longitudinal error of 3.7 cm and 5.2 cm, respectively, compared to ORB-SLAM2’s errors of 4.9 cm and 6.4 cm. The average and maximum static lateral errors were 3.6 cm and 6.1 cm, respectively, compared to ORB-SLAM2’s errors of 4.7 cm and 7.7 cm. Thus, the proposed fusion system improves the static positioning accuracy by 12 mm in the longitudinal direction and 11 mm in the lateral direction, effectively preventing positioning drift during the prolonged stationary periods of the rollers.

### 4.3. Straight-Line Compaction Test

#### 4.3.1. Short Straight-Line Compaction Positioning Test

To validate the positioning accuracy of the fusion positioning system for unmanned rollers under varying illumination intensities, experiments were conducted in both bright and dim environments. The roller was driven forward and backward over a total distance of 20 m, yielding positioning trajectories for both the fusion system and the traditional visual system, as shown in Figure 12 and Figure 13. In these figures, the red curve represents the actual compaction trajectory of the roller’s steel wheels. By comparing the positioning results of the fusion system and the traditional visual system with the actual compaction path, the real-time lateral errors were obtained, as shown in Figure 14a. The analysis of the real-time lateral errors during forward and backward compaction yielded the average and maximum lateral errors, as depicted in Figure 14b.

At an illumination intensity of 96 lux, the proposed fusion positioning system demonstrated slight improvements in both the average and maximum lateral errors, reducing them from 6.9 cm to 5.8 cm. At an illumination intensity of 18 lux, the visual positioning system based on ORB-SLAM2 exhibited multiple instances of abrupt positioning changes. This degradation was due to the poor illumination conditions, which hindered the clear extraction and tracking of the keypoints in the images, leading to decreased pose estimation accuracy and irreversible changes in the positioning accuracy.

By incorporating LiDAR odometry pose constraints, the fusion positioning system effectively mitigates the impact of poor illumination on the positioning accuracy, preventing sudden increases in errors and ensuring accurate short-distance linear compaction in tunnels. Specifically, under an illumination intensity of 18 lux, the average lateral error during forward compaction was reduced from 11.6 cm to 7.7 cm, and, during backward compaction, it was reduced from 11.3 cm to 7.4 cm. This demonstrates a significant enhancement in lateral positioning accuracy during short straight-line compaction.

#### 4.3.2. Long Straight-Line Compaction Positioning Test

Under different illuminations, a 40-m-long straight-line compaction positioning experiment was conducted with the roller moving forward and backward. The real-time positioning trajectories for different systems are illustrated in Figure 15 and Figure 16. The experiment sought to confirm whether long straight-line compaction impacted the positioning error of the fusion system. The real-time positioning errors under different illumination levels were derived from the actual roller trajectory, as presented in Figure 17a. Additionally, the average and maximum lateral errors are displayed in Figure 17b.

Under a dim environment of 18 lux, the ORB-SLAM2 system exhibited sudden error spikes in both forward and backward compaction, occurring six times. In contrast, the fusion positioning system displayed no such behavior. Furthermore, it reduced the average lateral error during forward compaction from 11.0 cm to 7.4 cm and during backward compaction from 10.5 cm to 7.0 cm. Thus, in dim environments, the proposed system enhances the lateral positioning accuracy in long straight-line compaction, unaffected by the increase in the compaction length.

As the linear positioning experiment involved both forward and backward compaction, the camera captured the ground image behind the roller. When the roller compacted forward, there were no compaction marks on the ground behind the roller. During backward compaction, the marks from the forward movement could be captured. This significantly increased the number of keypoints in the images, enhancing the accuracy of both matching and pose estimation. Consequently, in both Figure 14b and Figure 17b, the average error in the backward direction is observed to be smaller than that in the forward direction.

#### 4.3.3. Real-Time Performance of the Positioning System

During the short straight-line positioning experiment, the runtimes of various components within the fusion positioning system were recorded, including feature extraction and matching based on SURF operators, the improved loop closure detection method and range-based 2D LiDAR odometry. As shown in Figure 18, the entire process involved forward and reverse compaction, and the maximum, minimum and average runtimes for each step were recorded separately, as shown in Table 2. The visual feature extraction and matching based on SURF operators had an average runtime of 31.35 ms, making it the most time-consuming step due to the processing of a large number of feature points. Regarding loop closure detection, since keyframes are grouped, loop closure detection is temporarily suspended when moving forward; it begins only when the roller starts to move in the backward direction, with the similarity check of the keyframes, and the average runtime for the entire process is 4.72 ms. In this experiment, the 2D LiDAR with a smaller data volume was used, which collects data at a frequency of 15 Hz, the same as the camera, resulting in an average runtime of 7.92 ms. From this, it can be concluded that the total computational time of the above steps does not exceed the data reception frequency of the camera and LiDAR per unit time; therefore, the system can provide real-time and high-precision positioning.

### 4.4. Lane-Changing Positioning Test

Another crucial metric for the evaluation of fusion systems is the lane-changing positioning accuracy, directly impacting the compaction overlap width and preventing under-compaction or over-compaction. Thus, apart from linear positioning experiments, lane-changing compaction positioning experiments were conducted under different illumination levels for both the fusion system and the visual system. The positioning trajectories are depicted in Figure 19. The real-time lateral errors under different illumination levels are illustrated in Figure 20, with the average and maximum lateral errors tabulated in Table 3.

Under 101 lux illumination, the fusion positioning system exhibits a slight improvement in lateral error compared to the visual system. Meanwhile, at 17 lux illumination, the maximum lateral error decreases from 21.5 cm to 13.5 cm, and the average value decreases from 11.1 cm to 7.4 cm. Thus, during the lane-changing compaction of the unmanned roller, the fusion positioning system effectively mitigates the decrease in positioning accuracy attributed to poor illumination.

By integrating an enhanced loop closure detection method and LiDAR odometry with visual odometry, the fusion system achieves smaller positioning errors and better robustness. It addresses the challenge of decreased visual positioning accuracy resulting from changes in the illumination intensity within tunnels, as evidenced by the combined outcomes of the static, straight-line and lane-changing experiments.

## 5. Conclusions

In tunnel construction, fluctuations in illumination intensity can compromise the accuracy of visual positioning for unmanned rollers. To mitigate this challenge, this paper proposes an indoor fusion positioning system for unmanned rollers based on the camera/LiDAR. This system integrates loop closure detection and LiDAR odometry with the foundation of visual odometry. Keyframe poses serve as nodes, and constraints including the relative pose constraints between adjacent frames, loop closure constraints with historical frames and LiDAR odometry constraints are amalgamated through graph optimization. Given the prevalence of similar scenes in tunnels, a candidate frame selection method based on the compaction process is proposed to obtain precise loop closure constraints.

Through on-site experiments, compared with traditional loop closure detection methods, it is found that the improved method significantly enhances the precision while reducing the runtime, meeting the real-time system requirements. In the static positioning tests, the longitudinal/lateral accuracy of the roller indoor fusion positioning system improves by 12 mm and 11 mm, respectively. In the straight-line positioning tests under different illumination levels, the system’s positioning error is notably diminished. In two straight-line compaction tests with an illumination intensity below 20 lux, the average lateral error is increased by 34.1% and 32.8%, respectively. Furthermore, in the lane-changing positioning tests, this system boosts the positioning accuracy by 33% in dim environments. It enables substantial enhancements in positioning accuracy during straight-line and lane-changing compaction compared to visual positioning, effectively circumventing the declines in positioning accuracy due to illumination changes in tunnels.

Due to the constraints of laboratory environments, it is only feasible to simulate a limited range of illumination variations, such as 20 lux or 100 lux; rapid changes in illumination cannot be promptly mimicked. Additionally, this research builds on an existing visual positioning system by integrating 2D LiDAR. Future studies will need to incorporate more sensors, such as millimeter-wave radar and IMUs, to determine whether integrating additional sensors enhances the accuracy and robustness of the positioning system over the current configuration. Lastly, while the method proposed in this paper focuses on the impact of lighting variations in tunnel environments on the positioning accuracy, it does not address the performance of road rollers in outdoor settings. Future work will therefore extend the analysis and testing to outdoor scenarios, such as highways, to improve the versatility and environmental adaptability of the positioning system.

## Figures and Tables

**Figure 1 sensors-24-04408-f001:**
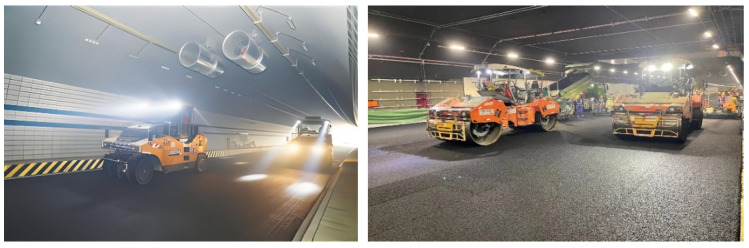
Tunnel compaction scenes.

**Figure 2 sensors-24-04408-f002:**
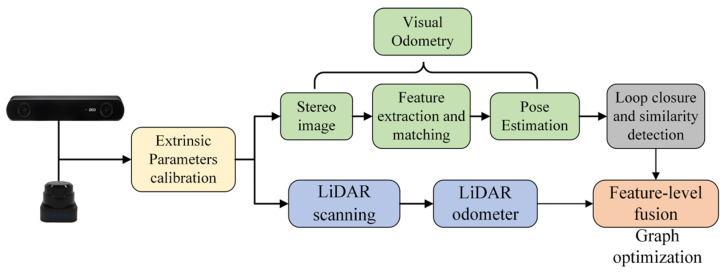
System framework diagram.

**Figure 3 sensors-24-04408-f003:**

An example of visual odometry.

**Figure 4 sensors-24-04408-f004:**
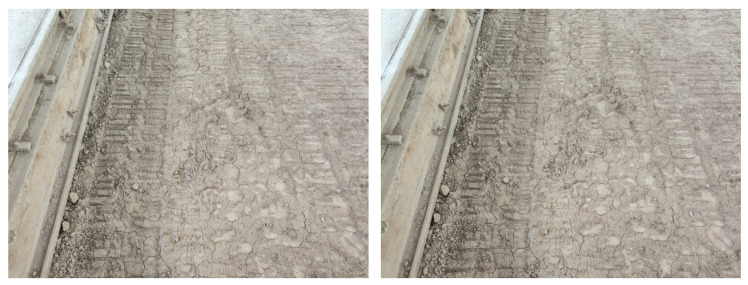
Error loop closure frames in similar scenes.

**Figure 5 sensors-24-04408-f005:**
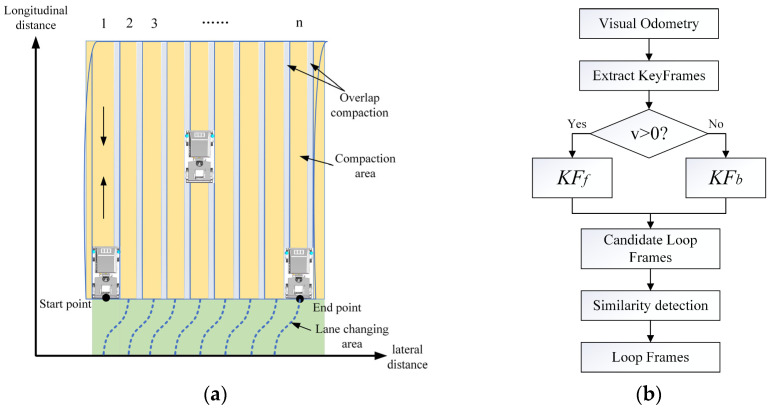
Schematic diagram of loop detection based on compaction process: (**a**) roller construction process; (**b**) loop closure detection process.

**Figure 6 sensors-24-04408-f006:**
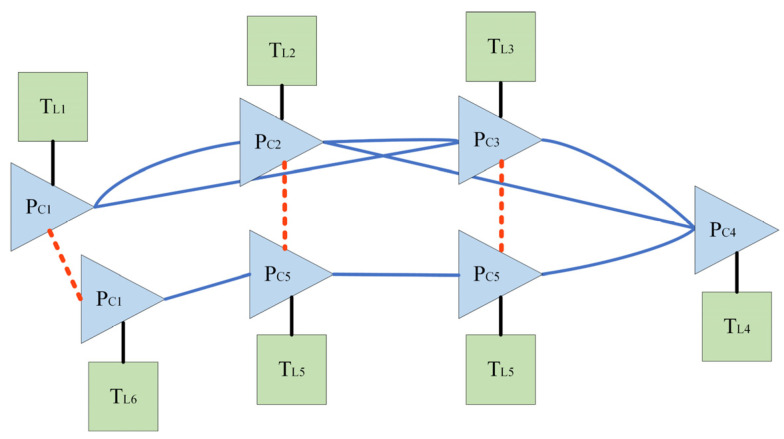
Fusion model based on graph optimization.

**Figure 7 sensors-24-04408-f007:**
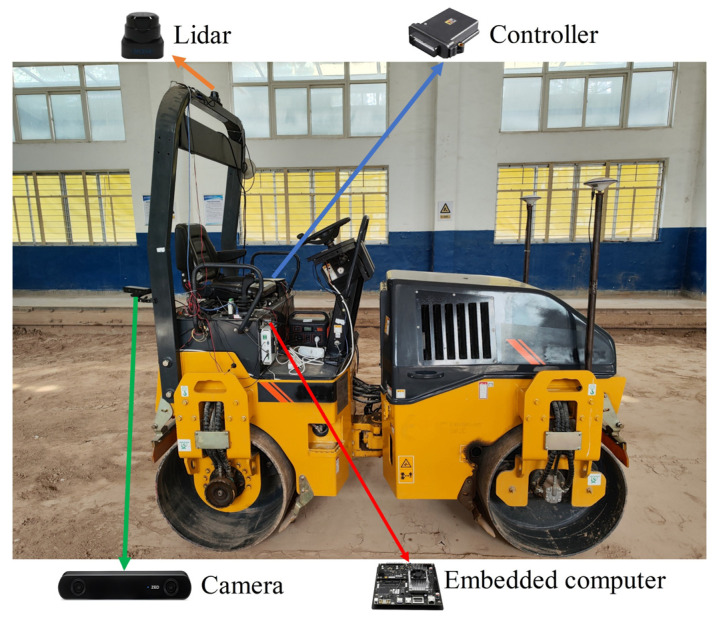
Distribution of sensors.

**Figure 8 sensors-24-04408-f008:**
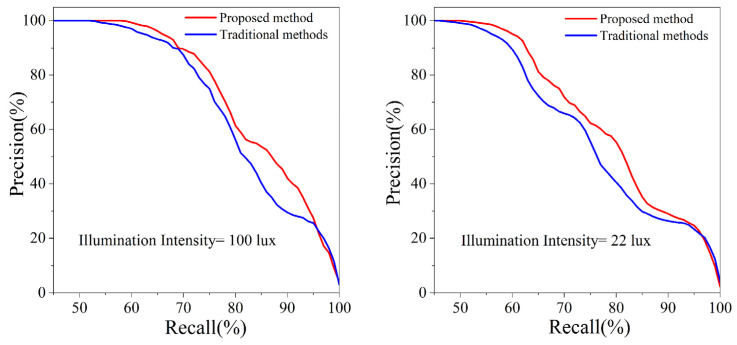
Precision–recall curve under different illumination levels.

**Figure 9 sensors-24-04408-f009:**
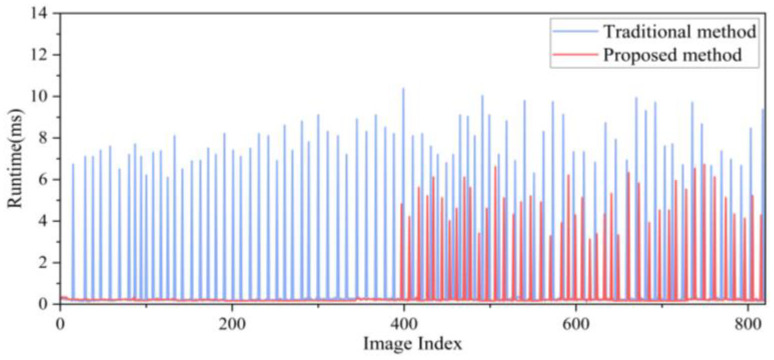
Running time for loop closure detection.

**Figure 10 sensors-24-04408-f010:**
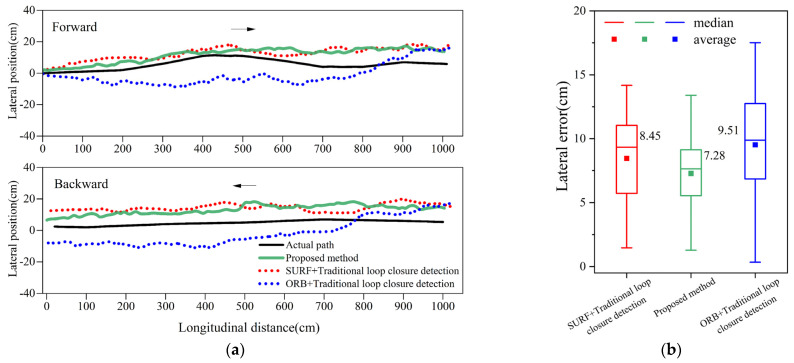
Positioning experiment with different loop closure detection methods: (**a**) positioning trajectory; (**b**) positioning error.

**Figure 11 sensors-24-04408-f011:**
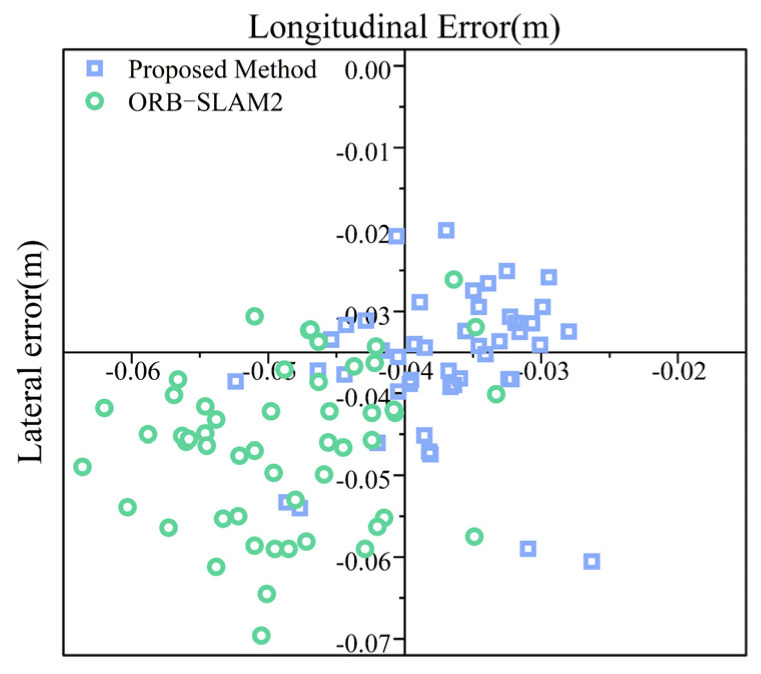
Static positioning error diagram.

**Figure 12 sensors-24-04408-f012:**
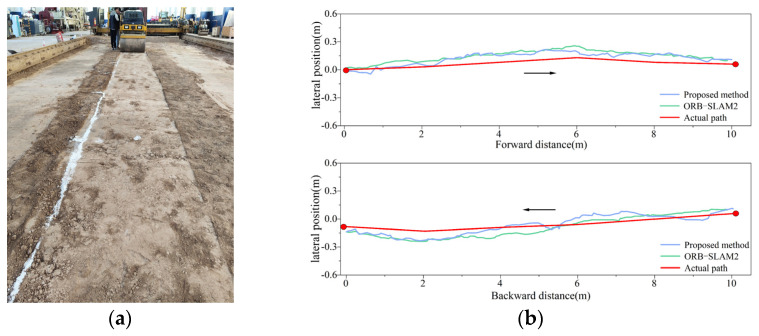
Bright environment, illumination intensity = 96 lux: (**a**) test site; (**b**) positioning data.

**Figure 13 sensors-24-04408-f013:**
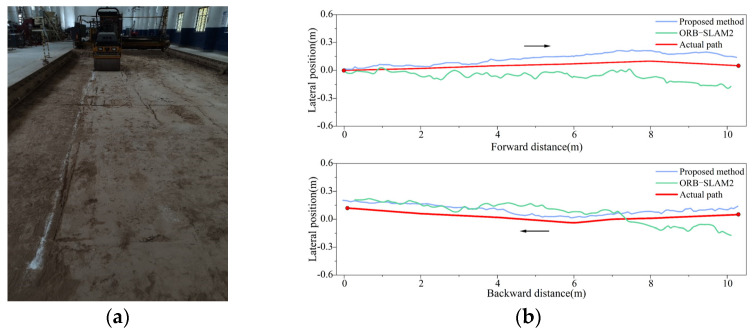
Dim environment, illumination intensity = 18 lux: (**a**) test site; (**b**) positioning data.

**Figure 14 sensors-24-04408-f014:**
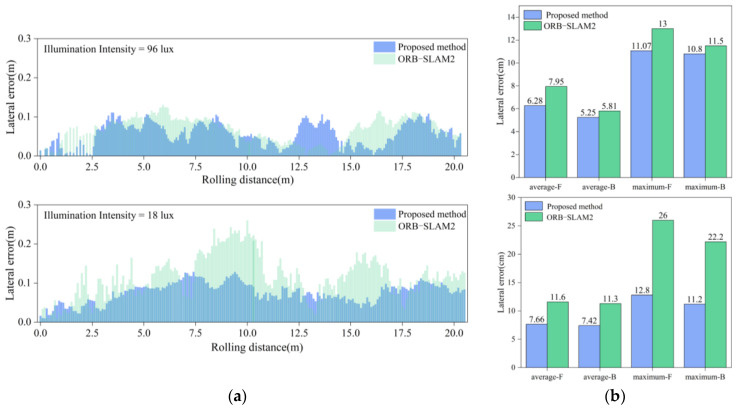
Lateral error under different illumination levels: (**a**) real-time lateral error; (**b**) average and maximum lateral error.

**Figure 15 sensors-24-04408-f015:**
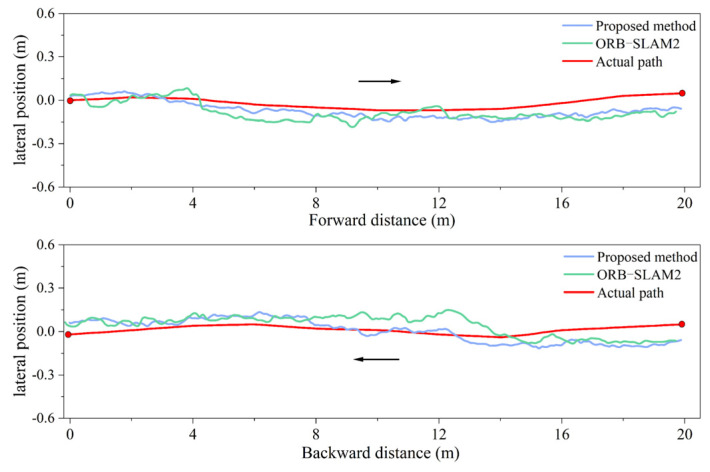
Forward and backward trajectory at illumination intensity of 95 lux.

**Figure 16 sensors-24-04408-f016:**
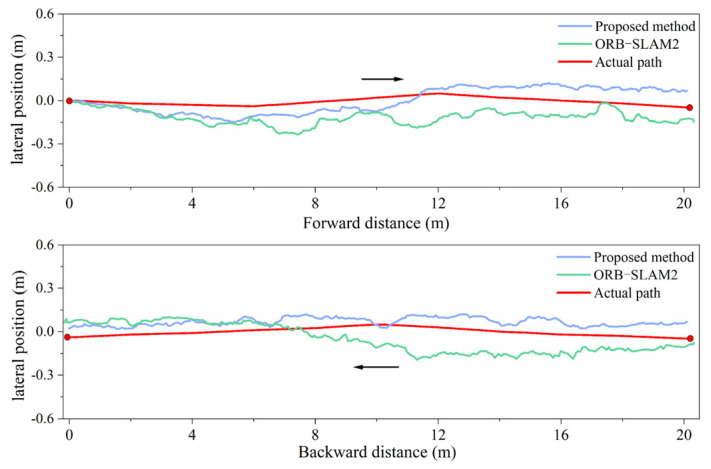
Forward and backward trajectory at illumination intensity of 20 lux.

**Figure 17 sensors-24-04408-f017:**
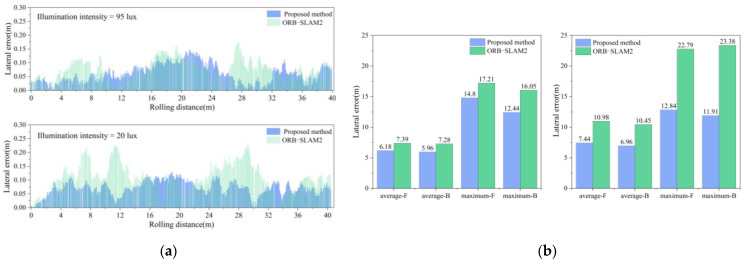
Lateral error under different illumination levels: (**a**) real-time lateral error; (**b**) average and maximum lateral error.

**Figure 18 sensors-24-04408-f018:**
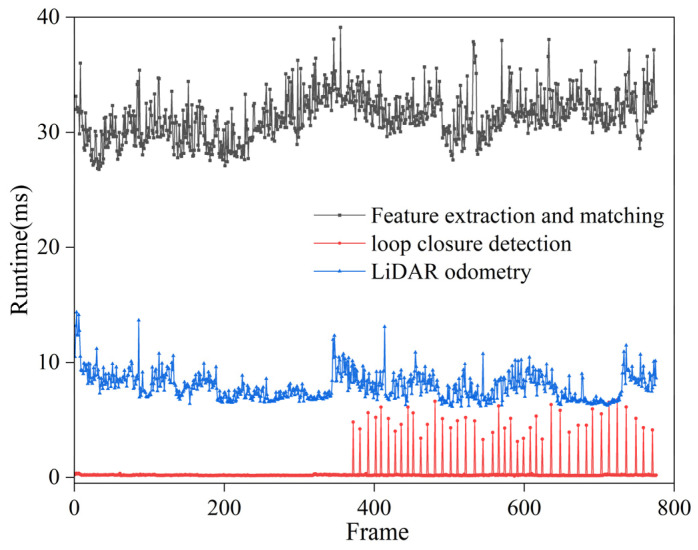
Graph of computational time variations for short straight-line positioning experiment.

**Figure 19 sensors-24-04408-f019:**
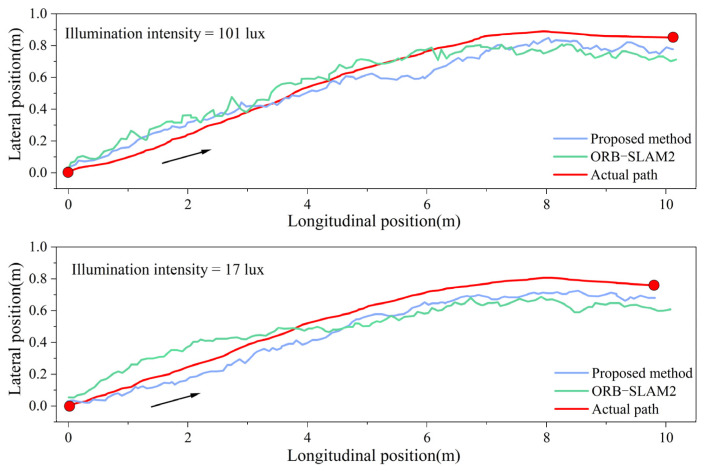
Lane-changing positioning data under different illumination.

**Figure 20 sensors-24-04408-f020:**
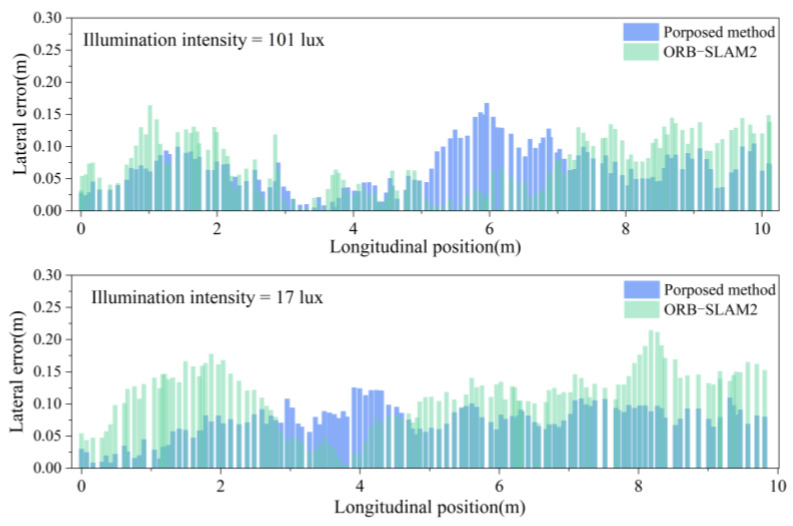
Lane-changing positioning error under different illumination.

**Table 1 sensors-24-04408-t001:** Parameters of double-drum roller.

Parameter	Value
Compaction width	1200 mm
Maximum steering angle	±20°
Rolling speed	2–5 km/h

**Table 2 sensors-24-04408-t002:** Calculation schedule for each step.

Parameter	Minimum (ms)	Maximum (ms)	Mean (ms)
Feature extraction and matching	26.77	39.12	31.35
Loop closure detection	0.14	6.73	4.72
2D LiDAR odometry	6.14	14.36	7.92

**Table 3 sensors-24-04408-t003:** Lane-changing positioning errors.

	Illumination Intensity = 101 lux		Illumination Intensity = 17 lux	
	Average Lateral Error (cm)	Maximum Lateral Error (cm)	Average Lateral Error (cm)	Maximum Lateral Error (cm)
ORB−SLAM2	7.2	16.4	11.1	21.5
Fusion System	6.6	16.8	7.4	13.5

## Data Availability

Data sharing is not applicable.

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
