# Peer review of "A Fusion Positioning System Based on Camera and LiDAR for Unmanned Rollers in Tunnel Construction"

_sensors, 2024, doi:10.3390/s24134408_

Round 1

Reviewer 1 Report

Comments and Suggestions for Authors

This manuscript presents a fusion positioning system based on camera and LiDAR for unmanned rollers in tunnel construction. This is an interesting topic since rollers are a type of users of unmanned systems. However, overall, the method is acceptable, but it is not well organized, especially the sections on methodology and experiments.

1. More details on previous work should be added to highlight the novelty of your approach. For instance, the limitations of the methods that are used in GNSS-denied tunnel environments, especially for the rollers.

2. L65. I cannot agree with the word "indoor". I think GNSS-denied could be more precise.

3. The presented approach should be compared with both traditional visual-based and LiDAR-based approaches, rather than only with a traditional visual-based one.

4. The resolutions of the figures should be increased to 600×600dpi.

5. The structure of the Section 3 should be adjusted. For now, it shows that the proposed method includes five steps/modules, but such things cannot be matched to Figure 2. Moreover, the following sub-sections are also not aligned to them. More importantly, things like Evaluation Indicators are not parts of the method.

6. The traditional methods (in Figures 8 and 9) in Section 4 are unclear. The parameters and processes of the traditional methods should be presented, as they are the baselines of the tests.

7. L388. The ORB-SLAM2 is suddenly employed for comparison. My question is why this SLAM is selected? As we all know, many SLAMs can be deployed directly. Please also refer to my comment 3.

8. Section 4 should be reduced and re-organized. In this version, it is too long compared to other sections.

Author Response

Thank you for all your suggestions. I have made revisions to the manuscript and responded to each of your suggestions.

Reviewer 2 Report

Comments and Suggestions for Authors In this manuscript, the author first emphasizes that traditional roller construction methods cannot meet the high-quality and high-efficiency requirements for subgrade and pavement construction, nor can they ensure the safety of personnel on the construction site. This paper proposes an unmanned roller positioning system based on the fusion of cameras and LiDAR, aiming to address the issues of GNSS positioning failure and the decline in visual positioning accuracy under varying lighting conditions during tunnel construction. By introducing loop closure detection and LiDAR odometry, the paper demonstrates the superiority of this system in tunnel environments. The following issues are identified: 1. The related work section (Section 2) of the article only briefly introduces some existing positioning methods, and the comparative analysis is not in-depth enough. It is recommended to include a more systematic review of existing studies, particularly regarding the effectiveness and limitations of camera and LiDAR fusion positioning technologies in different application scenarios. 2. The methodology section (Section 3) describes the system framework and each module in a rather general manner, lacking specific implementation details. For example, the feature extraction and matching algorithms of the visual odometry, the point cloud matching algorithms of the LiDAR odometry, and the specific implementation steps of the graph optimization need to be detailed. 3. In Section 3.2.1, although the innovation in the loop closure detection method (candidate frame selection based on the compaction process) is mentioned, the specific implementation process and algorithm principles need to be further elaborated. 4. The paper does not include a performance analysis of the proposed algorithms in terms of computational complexity and runtime. 5. In the experimental section 4.1.1, satisfactory results were obtained under varying lighting conditions of 100 lux and 22 lux. However, in practical applications, lighting conditions are often unstable.     6. The discussion section (Section 5) primarily focuses on summarizing the experimental results and does not delve deeply into discussing the limitations of the system and future research directions. It is recommended to include a discussion on the limitations of the system in practical applications, such as sensor accuracy, computational complexity, environmental adaptability, etc., as well as possible directions for improvement.

Author Response

Thank you for all your suggestions. I have made revisions to the manuscript and responded to each of your suggestions

Round 2

Reviewer 1 Report

Comments and Suggestions for Authors

All my comments have been replied and reflected.

Author Response

Thank you for your review.

Reviewer 2 Report

Comments and Suggestions for Authors

I accept the current version. 

Author Response

Thank you for your review.